# END-TO-END LEARNING OF VIDEO COMPRESSION USING SPATIO-TEMPORAL AUTOENCODERS

## ABSTRACT

*Deep learning* (DL) is having a revolutionary impact in image processing, with DL-based approaches now holding the state of the art in many tasks, including image compression. However, video compression has so far resisted the DL revolution, with the very few proposed approaches being based on complex and impractical architectures with multiple networks. This paper proposes what we believe is the first approach to end-to-end learning of a single network for video compression. We tackle the problem in a novel way, avoiding explicit motion estimation/prediction, by formalizing it as the rate-distortion optimization of a single spatio-temporal autoencoder; *i.e.*, we jointly learn a latent-space projection transform and a synthesis transform for low bitrate video compression. The quantizer uses a rounding scheme, which is relaxed during training, and an entropy estimation technique to enforce an information bottleneck, inspired by recent advances in image compression. We compare the obtained video compression networks with standard widely-used codecs, showing better performance than the MPEG-4 standard, being competitive with H.264/AVC for low bitrates.

## 1 INTRODUCTION

The fast technological improvements in imaging devices lead to higher-resolution media, outpacing the increase in capacity to store and transfer this data. Consequently, there is a growing need for more efficient (preferably patent-free) image and video compression schemes to expedite the process of sharing visual media over limited bandwidth channels, including the Internet.

The strong influence of DL in image processing coupled with the aforementioned relevance of image compression have sparked a quest for DL-based approaches, which have recently achieved results on par with state-of-the-art codecs (*e.g.*, WebP and BPG) (Ballé et al., 2018; Rippel & Bourdev, 2017). Different architectures have been proposed, generally based on a common principle: the original image is transformed into a latent representation by an encoder; this representation is quantized and entropy-coded; a decoder reconstructs the image from the quantized latent representation. This type of approach relies on the ability of *deep neural networks* (DNN) to extract meaningful and compact representations from 2D data. Convolutional autoencoders are particularly suited to this task, due to their ability to exploit the structural redundancy present in images and learn efficient representations.

In contrast with image compression, DL-based video compression methods are very scarce. As far as we know, only a few proposals have been published: compression of key frames followed by interpolation-based reconstruction of the other frames (Wu et al., 2018); block-based frame prediction combined with frame reconstruction (Chen et al., 2018; 2017). Those methods use complex multi-network architectures inspired by traditional video compression schemes. In this work, we depart from the traditional video coded structures, by avoiding any type of explicit motion prediction and tackling video compression via end-to-end optimization of a single spatio-temporal autoencoder architecture. Our approach is partly motivated by previous work on spatio-temporal and 3D autoencoders by (Hara et al., 2017; Zhu et al., 2014; Qiu et al., 2017). Just as image compression uses a projection transform to extract spatial information, a projection transform that analyzes the spatial-temporal redundancy within and between frames is able to learn a meaningful latent-space representation of video. Hence, we build upon state-of-the-art DL-based image compression, extending it from the 2D realm of images to the 3D world of video data.

We propose a rate-distortion optimization framework for video compression that also enforces temporal consistency between frames, and a spatio-temporal autoencoder architecture using three dimensional convolutions. Inspired by and extending recent work in image compression (Theis et al., 2017; Ballé et al., 2016; Ballé et al., 2018), we perform end-to-end learning of an autoencoder, by optimizing a loss function that combines: reconstruction distortion; an estimate of the length of the entropy-coded quantized latent representation; a temporal consistency loss. Although the resulting video compression network is not competitive with state-of-the-art codecs, namely the highly developed and optimized H.265/HEVC (Sullivan et al., 2012), at low bitrates, it is competitive with the previous-generation (also highly optimized) H.264/AVC and outperforms MPEG-4. The contributions of this paper can be summarized as follows:

- a simple formulation of the video compression problem as the search for a latent-space representation, from which the video is reconstructed after quantization;

- an end-to-end rate-distortion optimization framework adapted for video, which minimizes inconsistencies between frames;

- a 3D convolutional autoencoder architecture, with multi-scale connections applied to video compression, trained using the aforementioned optimization framework;

- an extension of the hyperprior proposed by Ballé et al. (2018) to a spatio-temporal application, specifically tailored to exploit remaining temporal information.

## 2 RELATED WORK

DL-based image compression was spotlighted by Toderici et al. (2015) and Toderici et al. (2016). Although not initially competitive with classical methods (JPEG and JPEG2000), these recurrent neural networks architectures already consisted of an encoder for extracting a latent representation, a bottleneck and quantization strategy, and a decoder to reconstruct the image. Lossy compression was enforced by the quantization step and by reducing the dimensionality of the latent representation at the bottleneck layer. However, the lack of explicit ways to constrain the amount of information in the latent representation, made it difficult to learn more efficient representations.

To circumvent this, Ballé et al. (2016) and Theis et al. (2017) optimize a transform for rate-distortion performance, where a quantized latent representation is modeled by a probability distribution, which is used to enforce a decrease in information entropy. Ballé et al. (2016) further show that the rate-distortion optimization of a compressive autoencoder is formally equivalent to certain variational autoencoder formulations (Kingma & Welling, 2013). Rippel & Bourdev (2017) focused on reconstruction and adversarial training instead of rate-distortion optimization; they achieve higher perceptual quality at low bitrates. However, the adversarial training is difficult to stabilize and introduced hallucinatory artifacts, which are strongly undesirable for compression tasks.

Very recently, Ballé et al. (2018) extended the rate-distortion optimization approach. They propose a new density model and fit a Gaussian distribution to the latent representation, whose parameters are sent in the bitstream as side-information. Results are comparable to state-of-the-art methods such as Google's *WebP* and *BPG* (`bellard.org/bpg/`), or better when using perceptual quality metrics.

Although video compression can be seen as a natural extension to these works, state-of-the-art DNN architectures for image compression are not directly applicable to higher-dimensional data such as video. Hence, Wu et al. (2018) address video compression as an interpolation task using two networks: one for image compression; another for interpolation, guided by optical flow or block motion information. The idea is to exploit temporal redundancy between consecutive frames by learning a continuity model for interpolation. Although this method yields results almost comparable to H.264/AVC, it has a significant computational overhead. In contrast, Chen et al. (2018) exploit temporal information through a voxel-based prediction module, also achieving performance comparable to H.264/AVC. Their prediction module uses a spatio-temporal dependency model to predict subsequent voxels, with the residuals being then stored for transmission. However, prediction must be used during both encoding and decoding, involving a significant computational burden.

A natural way of avoiding the hand-crafted nature, and computational burden, of frame prediction or motion estimation is to directly design and optimize a network for efficient video representation, as is done for image compression. The network can therefore be divided into two transformations: one

for projection and one for synthesis. For this, spatio-temporal autoencoders are a sensible choice, as they extend the convolutional autoencoder formulation with the ability to extract information about spatial and temporal dependencies in the data. *Convolutional recurrent neural networks* (CRNNs) are one possible formulation to combine the extraction of spatial information using convolutions (Shi et al., 2015), with the state propagation between iterations allowing to exploit temporal dependencies. CRNNs have been successfully used for tasks requiring spatio-temporal reasoning (*e.g.*, in the work of Wang et al. (2017); Liu et al. (2016); Patraucean et al. (2015)). However, CRNNs extract a latent representation for every frame, which is not efficient at low bitrates. Also it results in long run times for both encoding and decoding. The other option for spatio-temporal autoencoders is to use 3D convolutional neural networks (3DCNNs), which are structurally similar to convolutional autoencoders, but use 3D convolutions at each layer. This type of network is typically applied to problems in which the data resides on a 3D grid structure (*e.g.*, point clouds, 3D meshes or 3D medical imaging). 3DCNNs can also be adapted to process spatio-temporal data (*e.g.*, Zhao et al. (2017)) by organizing data on a 3D grid structure, allowing the joint processing of spatial and temporal information. However, 3DCNNs still require extensive research to produce efficient representations for data with as much redundancy as video.

## 3    PROPOSED APPROACH

As motivated and discussed above, we tackle video compression by optimizing a spatio-temporal autoencoder to learn an efficient representation of the input video. To that end, we formulate video compression as the problem of learning a low-entropy latent-space representation from which the original video can be reconstructed as well as possible. The compression loss results from quantizing the latent-space representation, thus discarding some information. By learning a probability model for the quantized latent representation, we are able to quantify the amount of information that the network uses for the quantized representation, and impose a limit on it. Since the reconstruction quality increases if more information is allowed to flow through the bottleneck, we attain a rate-distortion trade-off, as is common in compression algorithms. The proposed compression architecture uses two transforms: a projection transform that represents the original video in a latent space, and a synthesis transform that reconstructs the video in the original space. The video is processed on a 3D pixel grid with three input channels (*e.g.*, RGB or YUV). The compressed video is the result of entropy-coding the quantized latent-space representation, and the reconstruction is obtained by applying the synthesis transform to that representation. The transforms and the quantizer are jointly learned on an end-to-end fashion. Next, we begin by formally introducing the problem, then the proposed architecture, and finally we discuss the implementation.

### 3.1    PROBLEM FORMULATION

Formally, a video is a sequence of $N$ frames $f = (f_1, f_2, \ldots, f_N) \in \mathcal{F}^N$, where $f_i \in \mathcal{F}$, with $\mathcal{F}$ denoting the space to which each frame belongs (*e.g.*, $[0; R]^{W \times H \times 3}$, for RGB frames with $(W \times H)$ pixels, and a maximum pixel value of $R$). An encoder $\mathrm{E}_\phi : \mathcal{F}^N \to \mathcal{Z}$, where $\mathcal{Z} = \mathbb{R}^L$ is the latent space and $L$ its dimensionality, extracts a representation $z = \mathrm{E}_\phi(f)$. The latent-space representation is then quantized, $\bar{z} = q(z)$, where $q : \mathcal{Z} \to \mathcal{S}$ is a quantizer, with some finite code-book $\mathcal{S}$. Finally, the quantized latent-space representation $\bar{z} \in \mathcal{S}$ is used by a decoder $\mathrm{D}_\theta : \mathcal{S} \to \mathcal{F}^N$ to reconstruct an approximation $\hat{f}$ of the original video $f$.

In this formulation, $\phi$ and $\theta$ are the parameters of the encoder (or projection transform) and the decoder (synthesis transform), respectively. The whole network $\mathrm{C}_{\phi,\theta} : \mathcal{F}^N \to \mathcal{F}^N$ is the composition of the encoder, quantizer, and decoder:

$$\hat{f} = \mathrm{C}_{\phi,\theta}(f) = \mathrm{D}_\theta\Big(q\big(\mathrm{E}_\phi(f)\big)\Big). \tag{1}$$

In this work the quantizer $q$ represents quantization by rounding each component of $z$ to the nearest integer, thus $\mathcal{S}$ is simply a finite subset of $\mathbb{Z}^L$ (vectors of integers).

The compressed representation of the video results from the entropic coding of $\bar{z}$, which is a lossless operation, thus omitted in (1). Optimal entropic coding depends on the probability distribution of $\bar{z}$, $p_{\bar{z}}$, which must be estimated during training to allow the rate-distortion optimization of the network. For this we use a method inspired by the work of Ballé et al. (2018) and by noting that the

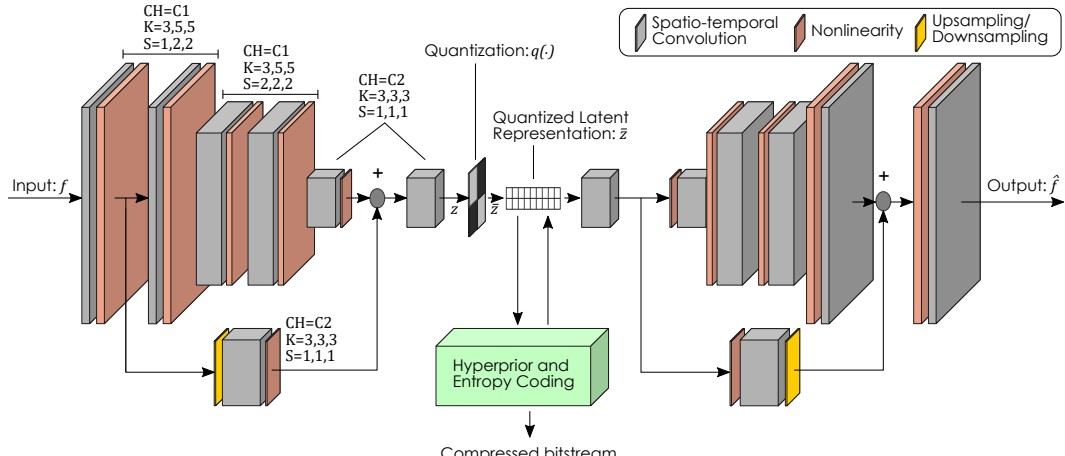

Figure 1: Illustration of the spatio-temporal autoencoder architecture. The parameters of each block correspond to the convolution configuration (CH – convolution channels; K – kernel size; S – stride). When the parameters are presented as a set of three numbers, the first refers to the temporal dimension and the remaining two to the spatial dimensions. Blocks with different sizes represent different hyper-parameter combinations.

components of $z$ are not independent. Each element of $z$, say $z_i$, is modeled by a zero-mean Gaussian distribution with variance $\sigma_i^2$, which should be dependent on other (in some neighbourhood) components of $z$. These variances are estimated by an additional autoencoder, $H_{\epsilon,\rho} : \mathcal{Z} \to \mathbb{R}_+^L$ (the *hyperprior* network), itself the composition of an encoder $X_\epsilon : \mathcal{Z} \to \mathcal{W}$ (where $\mathcal{W}$ is the latent space of the hyperprior), the same rounding quantizer $q$ as above, and a decoder $Y_\rho : \mathcal{S} \to \mathbb{R}_+^L$. As the variances are needed to decode the entropy-coded bitstream, the quantized latent representation of the hyperprior, $\bar{w} = q(w) = q\big(X_\epsilon(z)\big)$, must also be sent to the decoder as part of the compressed representation. Thus, the hyperprior can be seen as a side-information channel that improves the overall rate-distortion performance of the encoder, although contributing to the bitstream size.

To jointly optimize (in the rate-distortion sense) the main network $C_{\phi,\theta}$ and the hypeprior network $H_{\epsilon,\rho}$, an entropy model for the hyperprior needs to be estimated during training. The probability distribution of $\bar{w}$ is modeled as a factorized model

$$p(\bar{w}|\delta) = \prod_i p(\bar{w}_i|\delta_i), \tag{2}$$

where each $\delta_i$ is estimated during training. More specifically, each $\delta_i$ is obtained by approximating $p(\bar{w}_i|\delta_i)$ by a probability density function (pdf) defined in terms of its cumulative density function (cdf) $P : \mathbb{R} \to [0,1]$ (details given in the Appendix A). To simplify the model, we assume that $\delta_i = \delta_j$, if $i$ and $j$ are in the same channel of the hyperprior latent representation.

### 3.2 NETWORK ARCHITECTURE

The encoder $E_\phi$ is composed of processing blocks on two different scales (see Figure 1), for which we discuss the motivation in the next paragraph. Each processing block is composed of a 3D convolution and a non-linearity. Leaky ReLUs (Xu et al., 2015) with a leak of $0.2$ were used as the non-linearity. The latent representation results from adding the two scales and processing the result with a final inter-scale block. The shallower path uses trilinear downsampling to align both outputs. The decoder $D_\theta$ is built by reversing the order of the operations in the encoder, replacing the convolutions by their transposes, and downsampling operations by the corresponding upsamplings. The architecture's two initial blocks only use non-unit stride in the spatial dimensions, so as to apply a lower reduction in size on the temporal dimension than on the spatial ones.

Because dependencies exist in the data at different spatial scales, combining the results from different layers of a DNN has shown considerable improvements in many tasks, with residual networks

being an instance of this idea (He et al., 2015; Feichtenhofer et al., 2016). Temporal dependencies may also be present in short or long frame sequences (short or long time-scales). Motivated by this observation, we use a dual-scale architecture (see Figure 1) to better adapt to the variable dependency/redundancy scale of the input video. The main path of the network is structured as a regular deep encoder, while the secondary path resembles a high-level skip connection in some residual architectures. Notice that the secondary path is fed before applying any form of temporal compression. The two paths aggregate information from different temporal and spatial scales, yielding a more expressive latent representation. We evaluated the use of additional scales, but the improvements were marginal or non-existent; the increase in computational and memory requirements of additional scales makes their inclusion impractical.

Compared to the original formulation of Ballé et al. (2018), we extend the hyperprior network to estimate the variance from both the remaining spatial and temporal redundancy in $z$, by structuring $H_{\epsilon, \rho}$ also as a spatio-temporal autoencoder. The hyperprior network is not shown in detail in Figure 1, to keep it from becoming too complex. The encoder $X_\epsilon$ uses three sequential processing blocks with a 3D convolution ($C1$ channels, kernel size 3 and stride (1,2,2)) and a ReLU non-linearity. The decoder $Y_\rho$ inverts $X_\epsilon$ by replacing convolutions by the corresponding transposed convolutions.

### 3.3 OPTIMIZATION FRAMEWORK

The process of obtaining the compressed representation of a video and recreating the video from this representation is expressed in (1). However, parameter learning cannot be carried out directly on this structure, since the quantizer would block the backpropagation of the gradients through the bottleneck, a fact that has been pointed out by several researchers (Ballé et al., 2018; Theis et al., 2017; Johnston et al., 2017; Toderici et al., 2016). One way to circumvent this difficulty exploits a classical result from quantization theory (Gersho & Gray, 1992): high-rate quantization noise/error is well approximated by additive noise, with uniform distribution in the quantization interval. During training, the explicit quantization is thus replaced by additive noise with a uniform density on the interval $\left[-\frac{1}{2}, \frac{1}{2}\right)$, since the quantizer $q$ simply rounds each component of its input to the nearest integer. This method has been proposed and successfully used in image compression by Toderici et al. (2016); Ballé et al. (2018).

The proposed network is end-to-end optimized for both the reconstruction loss ($\mathcal{L}_r$) and the total entropy of the representation ($\mathcal{L}_h$). Additionally, we include a third loss term encouraging temporal consistency between consecutive frames ($\mathcal{L}_t$). The final optimization loss function $\mathcal{L}$ is expressed as the weighted sum

$$\mathcal{L}(f, \hat{f}) = \mathcal{L}_r(f, \hat{f}) + \alpha \, \mathcal{L}_h(f) + \beta \, \mathcal{L}_t(f, \hat{f}), \tag{3}$$

with $\alpha$ and $\beta$ empirically determined to balance training stability, proximity to a target bitrate, reconstruction quality, and temporal consistency. We next describe each term in (3) in more detail.

#### 3.3.1 RECONSTRUCTION LOSS $\mathcal{L}_r$

The adopted reconstruction loss is simply the *mean squared error* (MSE),

$$\mathcal{L}_r(f, \hat{f}) = \text{MSE}(f, \hat{f}) = \frac{1}{N} \sum_{i=1}^{N} \|f_i - \hat{f}_i\|_2^2. \tag{4}$$

Optimizing the MSE is equivalent to optimizing the PSNR (*peak signal-to-noise ratio*), since $\text{PSNR}(f, \hat{f}) = 10 \log_{10}\left(R^2 / \text{MSE}(f, \hat{f})\right)$ (where $R$ is the range of the pixel values), which is the most common metric used in assessing video compressors.

It is possible and would make sense to optimize more sophisticated metrics that correlate better with (human) perceptual quality, such as the well-known MS-SSIM (Wang et al., 2003)). However, optimizing the PSNR helps to adequately compare the proposed network with other codecs that are also optimized for PSNR, which we believe is a more relevant goal at this point of the work.

#### 3.3.2 ENTROPY LOSS $\mathcal{L}_h$

The approximate probability models learned during training allow estimating the length of corresponding optimal codes for the quantized variables (Cover & Thomas, 2006). The length of the

optimal codeword for some particular $\hat{z}_i$ is approximately (*i.e.*, ignoring that it has to be integer and assuming that the probability mass function of the quantized variable is well approximated by the corresponding pdf at that value) equal to $-\log p(\hat{z}_i)$, where $p(\hat{z}_i)$ is approximated by a Gaussian distribution, as described in Section 3.1; similarly, the optimal codeword for some particular $\hat{w}_i$ has length $-\log p(\hat{w}_i|\delta_i)$, as given by (2). Consequently, the total number of bits used by optimal codes for $\bar{z}$ and $\bar{w}$, under these probability models, is approximately given by

$$L(\bar{z}, \bar{w}) \simeq \sum_i -\log_2 \mathcal{N}(\bar{z}_i|0, \sigma_i^2) + \sum_j -\log_2 p(\bar{w}_i|\delta_i), \qquad (5)$$

where $\mathcal{N}(x|\mu, \sigma^2)$ denotes a Gaussian pdf of mean $\mu$ and variance $\sigma^2$, computed at $x$. Finally, since both $\bar{z}$ and $\bar{w}$ are functions of $f$, we have

$$\mathcal{L}_h(f) = L\Big(q\big(E_\phi(f)\big), q\big(X_\epsilon(E_\phi(f))\big)\Big). \qquad (6)$$

By penalizing the entropy of the quantized representations at the bottleneck, rather than simply its dimensionality, we control the amount of information to be used for reconstruction, thus forcing the network to jointly optimize the reconstruction loss and the bitrate.

### 3.3.3 TEMPORAL CONSISTENCY LOSS $\mathcal{L}_t$

The generative nature of the spatio-temporal autoencoder may produce inconsistencies between frames, mainly due to projection of a quantized latent representation back to the original video space. At low bitrates, the reconstructed video relies heavily on prior information embodied in the reconstruction sub-network $D_\theta$, which may yield inconsistencies between consecutive frames, since the reconstruction loss used in training operates on a frame-by-frame fashion. Since the human visual system is very sensitive to disruptions to the temporal consistency/continuity between frames, this may cause a serious degradation of the perceptual quality of the reconstructed video sequence. To address this problem, we introduce a short-term temporal consistency loss, $\mathcal{L}_t$, inspired by the work of Lai et al. (2018). This short-term temporal loss is based on the warping error between each pair of subsequent input frames and the corresponding pair of output frames. The warping uses the optical flow between the two input frames, obtained by a pre-trained version of *Flownet 2.0* (Ilg et al., 2016). More specifically, the temporal loss is defined as

$$\mathcal{L}_t = \sum_{t=2}^{N} ||M_t \odot (\hat{f}_t - \tilde{f}_{t-1})||_2^2, \qquad (7)$$

where $\tilde{f}_{t-1}$ is the result of warping the frame $\hat{f}_t$ to time $t-1$ by using the estimated backwards optical flow from $f_t$ to $f_{t-1}$, $M_t$ is a binary occlusion mask excluding pixels that are not present in both $\hat{f}_t$ and $\tilde{f}_{t-1}$ (thus are not comparable), and $\odot$ denotes pixel-wise product,

## 4 TRAINING AND RESULTS

### 4.1 EXPERIMENTAL SETUP

Our network was trained using PyTorch (source code to be released after paper acceptance) and optimized for five target bitrates, each using a different parameter setting (see table in Figure 2). We use two filter configurations (parameters C1 and C2), one concentrated on very low bitrates (networks A-C), the other allowing additional information to improve the video quality (networks D-E). The training set is composed of $10^4$ *high definition* (HD) videos with at least 1080p resolution ($1920 \times 1080$), randomly chosen from the YouTube-8M dataset (Abu-El-Haija et al., 2016). Each video was downscaled to half of its original size in both spatial dimensions (to minimize any existing compression artifacts) and sliced into 32-frame sequences, from each of which a random $128 \times 128$ spatial crop was extracted. Each training iteration uses a batch of 4 of these cropped sequences. The learning rate was fixed at $10^{-4}$ until the 5000-th iteration, beyond which a 0.2 decay was used. Each network was trained until it reached stability in compressing a validation set of 10 videos also extracted from the YouTube-8M collection (disjoint from the training set), downscaled to $640 \times 360$.

Evaluation was carried out on a collection of 10 uncompressed 1080p videos from the MCL-V database (Lin et al., 2015). That database was created to assess streaming codecs, thus it includes

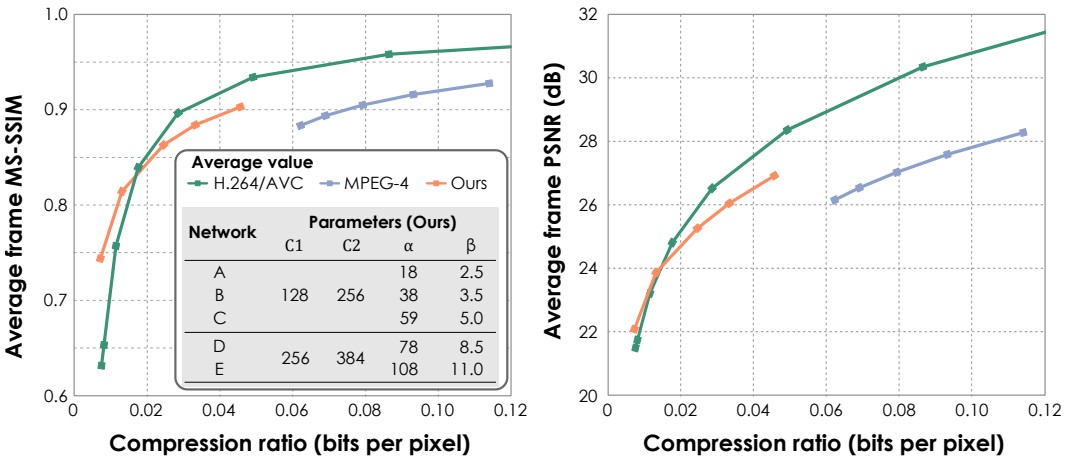

Figure 2: Summarized results for the MCL-V dataset for both MS-SSIM and PSNR.

a variety of scenes making it well-suited to showcases the general-purpose nature of the proposed framework; because the training set has very few synthetic videos, we have not included this type of videos in the test set. Each evaluation clip was downscaled to a width 640 pixels, maintaining its aspect ratio. Our architecture is evaluated against the MPEG-4 (Puri & Eleftheriadis, 1998) implementation in libxvid and the H.264/AVC (Wiegand et al., 2003) implementation in libx264. In both cases, the *ffmpeg* (www.ffmpeg.org) was used for compression. We used two different quality metrics: the classical PSNR, as well as MS-SSIM, which is often used to estimate the perceived quality of images and videos.

## 4.2 RESULTS

The average rate-distortion curve for each network and each profile level (MPEG-4 and H264/AVC) is shown in Figure 2. Individual curves for each of the test sequences are shown in Appendix B.

The results show that our video compression network attains similar quality levels than MPEG-4 in both metrics for significantly lower bitrates. Moreover, it has competitive performance to H.264/AVC at the low bitrates that are the focus of this paper. The artifacts produced by our network at low bitrates can be considered as perceptually more acceptable, compared to both H.264/AVC and MPEG-4, even at similar PSNR and MS-SSIM values. Figure 3 shows a comparison between a crop of two frames from the same evaluation clip at similar compression ratios. By relying on the learned latent space to perform video compression, the proposed DNN avoids the typical blocking artifacts of residual compression, conforming to the learned distribution of features and structure. In contrast, it generates natural looking artifacts, guaranteeing a more graceful degradation of quality for very low bitrates. In comparison, in this scenario, traditional codecs suffer from highly unnatural blocking artifacts. Although we are focused on the lower end of the bitrate axis, the diminishing returns of optimizing the network for higher bitrates in this architecture are noticeable. Our network is limited by the complexity of the latent space that the encoder and decoder can jointly learn; in turn they are limited by the expressiveness of both transformations. Increasing the number of filters allows, in principle, more complex models to be learned, which to some extent works, but also leads to an increase in computation cost and training time.

There is a particular situation in which DL-based compression struggles to achieve efficient bitrates for any compression rate: global motions (e.g., panning or zooming) in which consecutive frames are identical after a simple geometric transform. This is mostly due to the inability of the network to efficiently capture global transformations, a trivial task for conventional video codecs with simple hand-crafted heuristics and almost no cost in bitrate. This is one of the roadblocks that need to be resolved to further improve on the performance of this type of compression schemes. One possible approach, is to learn a form of spatial invariance that helps in abstracting global transforms between sequential frames.

Ours  (0.01702 BPP)                    H.264/AVC  (0.01739 BPP)

Figure 3: An example comparison between two compressed sequences (at similar compression ratios), one produced by our network, the other by H.264/AVC. A detailed zoom shows that distortions produced by our approach are more natural. The image can be zoomed to analyze other details.

Still, this works shows that through an end-to-end learning framework, with a single-network spatio-temporal auto-encoder, it is possible to achieve video compression performances for low bitrates that is competitive with classical codecs, designed with the contribution of several research groups over many years. Hence, this initial investigation should open the path for a new research line, focused on building efficient video compression networks while avoiding computational complex motion compensation and residual compression strategies.

## 5  CONCLUSION

While *deep learning* has raised the bar in many problems of image and video processing, including image compression, the same effect has not yet propagated to the area of video compression. Video is a complex type of data, for which the existing codecs are extremely well optimized, placing a very high barrier of entry into this field. In order to be competitive, the recently proposed DL-based methods rely on the same motion/frame prediction architecture as used in standard codecs, which eventually may lead to the same pitfalls. To push the research on DL-based video compression in a new direction, we first take a step back and abandon the established prediction-based architectures. Instead, the video compression problem is formulated as a latent space search within a limited entropy constraint. The problem can thus be approached with two transformations: one for projection to the learned latent space, and one for synthesizing the reconstructed video. The bitstream representing the compressed video results from entropy-coding a quantized version of its latent representation. We propose a rate-distortion optimization framework to train a single spatio-temporal autoencoder for both its reconstruction loss and entropy model. To combat temporal inconsistencies in the decoded videos, the optimization framework is augmented with a short-term temporal loss, encouraging the network to ensure temporal continuity between consecutive frames. Our work is, as far as we are aware, the first end-to-end learned video compression architecture using DL.

The obtained results show the potential of our network in low bitrates and that the introduced artifacts are more visually pleasing than the unnatural blocking artifacts of standard video codecs. There is still much to understand in this type of video compression, and so we did not expect to surpass the performance of the carefully-built standard video compression codecs. Instead, the focus was on demonstrating that end-to-end DL-based methods are a potential solution to video compression, which we believe was made clear in this work.

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

# A  Probabiliy Density Function

The pdf $p : \mathbb{R} \to \mathbb{R}^+$ in (2) is defined in terms of its cdf $P : \mathbb{R} \to [0, 1]$ $(P(x) = \int_{-\infty}^{x} p(y)dy)$, as a general approximator of density functions (Ballé et al., 2018). The two functions are defined as follows, for $0 < k \leq K$ (we set $K = 5$ in our architecture)

$$
\begin{aligned}
P &= g_K \circ g_{K-1} \cdots \circ g_1, \\
p &= g'_K g'_{K-1} \ldots f'_1, \\
g_k(x) &= h_k(H^{(k)}x + b^{(K)}), \\
g_K(x) &= \text{sigmoid}(H^{(k)}x + b^{(K)}), \\
h_k(x) &= x + a^{(k)} \odot \tanh(x).
\end{aligned} \tag{8}
$$

# B  DETAILED RESULTS

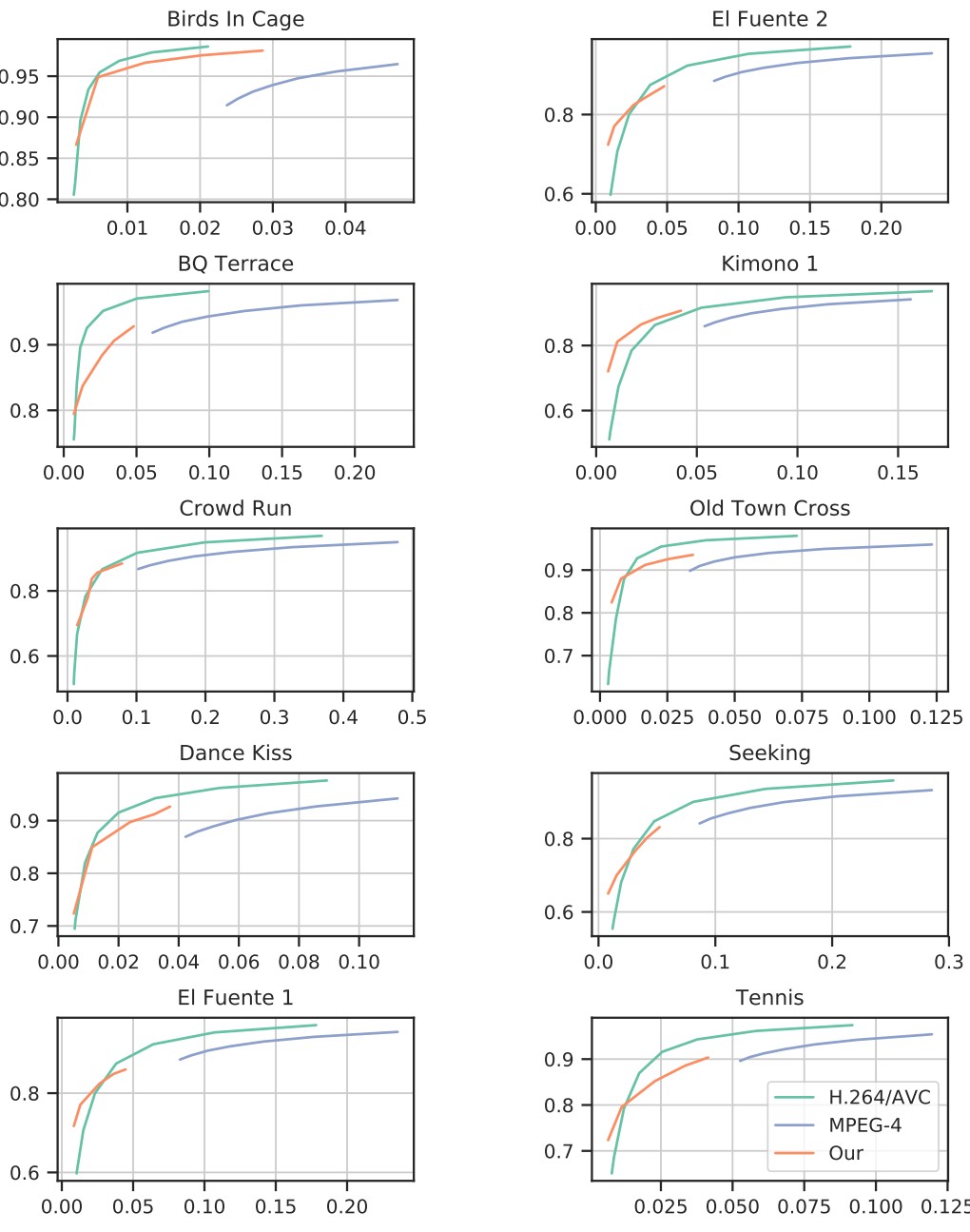

Figure 4: A grid of the individual MS-SSIM results for each one of the 10 considered video clips of the MCL-V dataset (with the same name as in the article for the dataset). The legend of the last graph applies to all the graphs in the grid.

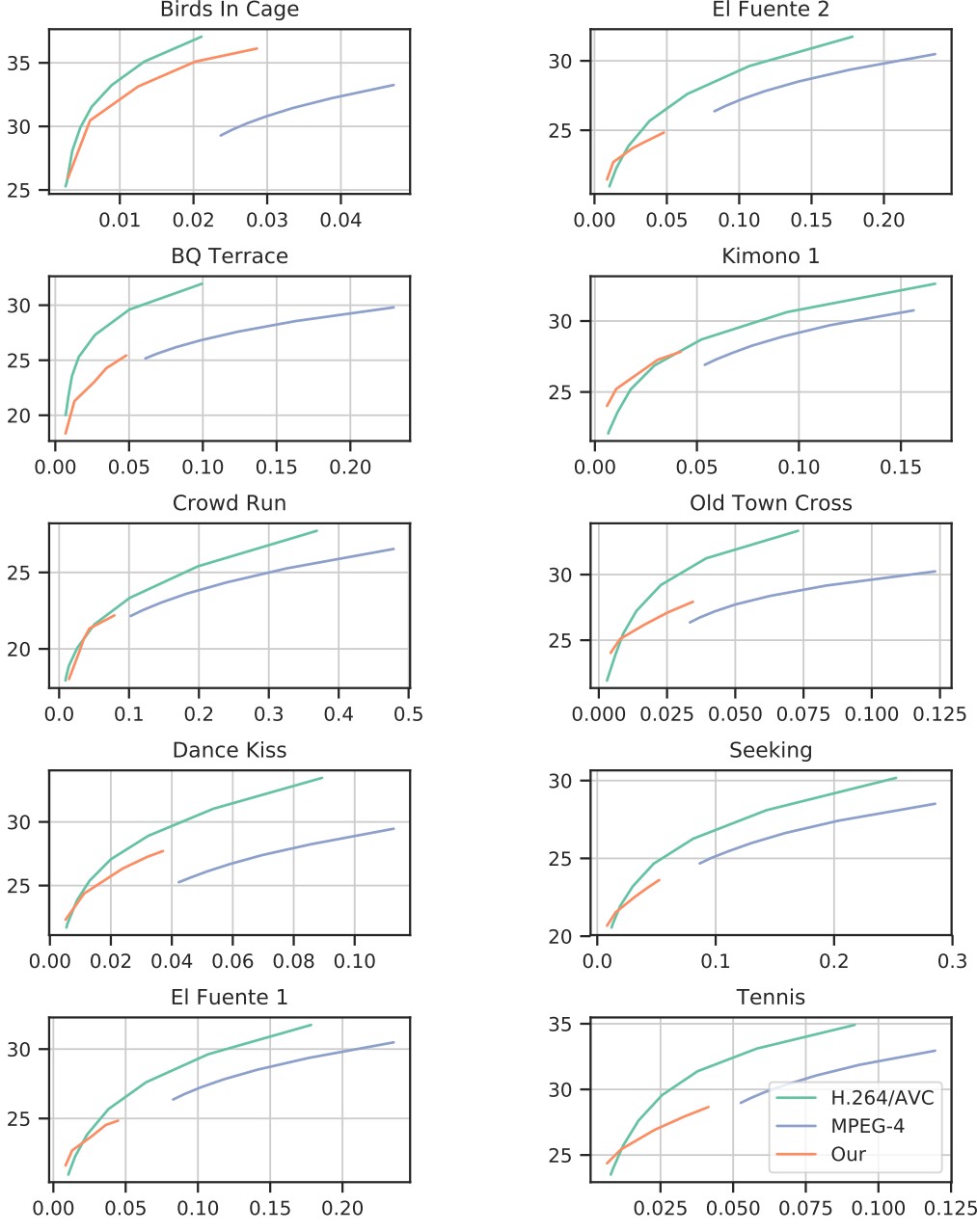

Figure 5: A grid of the individual PSNR (measured in dB) results for each one of the 10 considered video clips of the MCL-V dataset (with the same name as in the article for the dataset). The legend of the last graph applies to all the graphs in the grid.

