# OpenReview forum: "End-to-End Learning of Video Compression Using Spatio-Temporal Autoencoders"
_ICLR.cc/2019/Conference_

### Official Review · AnonReviewer2 · 2018-11-01
**marginally novel, evaluation is very flawed, and incomplete**

**Rating:** 2
**Confidence:** 5

**Review:**

First off, the paper presents a relatively straight-forward extension to video from the work done in image compression. The work uses 3D volumes instead of 2D images, and exploits this structure by adding a secondary network to both the encoder/decoder.

The work is therefore *marginally* novel, but it is one of the first to propose neural methods for compressing video.

My biggest complaint about this paper, however, is about evaluation. I don't think it's possible to take this paper seriously as is, due to the fact that the metrics use in the evaluation are absolutely skipped.

Given that this is such a crucial detail, I don't think we can accept this paper as is. The metrics need to be described in detail, and they should follow some previously used protocols (see below).

For example, in libvpx and libaom (which is the current best performing method for video compression - AV1), there are two versions of PSNR: Global and Average PSNR respectively, and this is what gets reported in publications/standards meetings.

Global PSNR: Compute MSE for the entire sequence combining Y, Cb, Cr components, and then compute PSNR based on the combined MSE.
Average PSNR: Compute MSE for each frame combining Y, Cb, Cr, components; then compute PSNR for the frame based on the combined MSE and cap it to a max of 100. Then average the PSNR over all the frames.

MPEG uses something like computing Average PSNR for each component (similar to what I mentioned above, but for each component) and then combine the Y-, Cb- and Cr- PSNRs using a weighted average. For 420 that will be equivalent to [4*MSE(y) + MSE(Cb) + MSE(Cr)/6. For 422 that will be equivalent to [2*MSE(y) + MSE(Cb) + MSE(Cr)/4. For 444 that will be equivalent to [MSE(y) + MSE(Cb) + MSE(Cr)/3.  Additionally, when using YCbCr, the authors also need to refer to which version of the color standard is employed, since there are multiple ITU recommendations, all of which differ in how to compute the color space transforms.

Please note that video codecs DO NOT OPTIMIZE FOR RGB reconstruction (humans are much more sensitive to brightness details than they are to subtle color changes), so comparing against them in that color space puts them at a distinct disadvantage. In the video compression literature NOBODY reports RGB reconstruction metrics.

Please note that I computed the PSNR (RGB) for H.264, on the resized MCL-V dataset (640x360) as the authors proposed and I observed that the metric has been ***MISREPRESENTED*** by up to 5dB. This is absolutely not OK because it makes the results presented not be trustworthy at all.

Here is the bpp/RGB PSNR that I obtained for H.264 (for completeness, this was computed as follows: used version 3.4.2 of ffmpeg, and the command line is "ffmpeg -i /tmp/test.y4m -c:v h264 -crf 51 -preset veryslow", tried many settings for crf  to be able to get roughly the same bpp per video, then compute RGB PSNR for each frame per video, aggregate over each video, then average cross videos):

BPP, Average PSNR RGB (again, not a metric I would like to see used, but for comparison's sake, I computed nonetheless -- also, note that these numbers should not be too far off from computing the average across all frames, since the video length is more or less the same)):
0.00719, 23.46
0.01321, 26.38
0.02033, 28.92
0.03285, 31.14
0.05455, 33.43

Similar comments go for MS-SSIM.

Lastly, it is unfair to compare against H263/4/5 unless the authors specify what profiles were used an what kind of bitrate targeting methods were used.

---

> ### Author Response · Authors · 2018-11-25
> **Response to reviewer comments**
>
> Thank you for your feedback and suggestions. We have double checked our results, and confirmed that the values on the graphs are correct according to the used method. However, we agree that it is necessary to properly clarify how the evaluation was done. Additionally, we also agree that it would make more sense to make the comparison on the YCbCr space, as it is common by the video coding community. We will take your input into consideration in a revised manuscript.

---

### Official Review · AnonReviewer1 · 2018-11-06

**Rating:** 3
**Confidence:** 4

**Review:**

This paper presents a spatiotemporal convolutional autoencoder trained for video compression. The basic model follows the logic of traditional autoencoders, with an intermediate quantizer:

input -> convolutional neural network with a skip connection as an encoder -> quantizer -> transposed convolutional neural network with a skip connection as a decoder.

As the quantizer is a non-differentiable operation, the paper proposes to follow (Toderici et al  2016, Balle et al, 2018) and cast quantization as adding uniform noise to the latent variables. The pre-quantized variables are modelled as Gaussians with variance that is predicted by a second "hyperprior" network dedicated to this task. The final model is trained to minimize three losses. The first loss minimizes the difference between the true frame pixel values and the predicted pixel values. The second loss minimizes the entropy of the latent codes. The third loss minimizes the difference between neighboring pixels in subsequent frames, ignoring those pixels that are not linked between frames. The model is trained on 10,000 videos from the Youtub-8M dataset and tested on 10 videos from the MCL-V database, with rather ok results.

Generally, parts of the proposed approach sound logical: an autoencoder like architecture makes sense for this problem. Also, the idea of using uniform noise to emulate quantization is interesting.  However, the paper has also weaknesses.

- The added novelty is limited and unclear.  Conceptually, the paper is overclaiming. Quoting verbatim from the conclusion: "Our work is, as far as we are aware, the first end-to-end learned video compression architecture using DL.", while already citing few works that also rely on deep networks (Wu et al., 2018, Chen et al., 2016). In the related work section it is noted that these works are computationally heavy. However, this doesn't mean they are not end-to-end. The claims appear to be contradicting.

- The technical novelty is also limited. What is new is the combination of existing components for the task of video compression. However, each component in isolation is not novel, or it is not explained as such.

- Parts of the model are unclear. How is the mask M computed in equation (7)? Is M literally the optical flow between frames? If yes, what is the percentage of pixels that is zeroed out? Furthermore, can one claim the model is fully end to end, since a non-differentiable optical flow algorithm is used?

- The purpose of the hyperprior network is unclear. Why not use a VAE that also returns the variance per data point?

- Most importantly, it is not clear whether the model is trained as a generative one, e.g., with using a variational framework to compute the approximate posterior. If the model is not generative, how can the model be used for generation? Isn't it then that the decoder simply works for reconstruction of already seen frames? Is there any guarantee that the model generalizes well to unknown inputs? The fact that the model is evaluated only on 10 video sequences does not help with convincing with the generalization.

- The evaluation is rather weak. The method is tested on a single, extremely small dataset of just 10 videos. In this small dataset the proposed method seems to perform worse in the majority of compression ratios (bits per pixel). The method does seem to perform a bit better on the very low bits per pixel regime. However, given the small size of the dataset, it is not clear whether these results suffice.

- Only two baselines are considered, both hand-crafted codecs: H.264/AVC and MPEG-4. However, in the related work section there are works that could also be applied to the task, e.g., the aforementioned ones. Why aren't these included in the comparison?

- Although it is explained that the focus is on the very low bitrates, it is not clear what part of the model is designed with that focus in mind. Is this just a statement just so to focus on the part of the curve in the experiment where the proposed method is better than the reported baselines? Is there some intrinsic model hypothesis that makes the model suitable for low bit rates?

In general, the paper needs to clarify the model and especially explain if it is (or not a generative one) and why. Also, a more extensitve arrays of experiments need to be executed to give a better outline of the methods capabilities and limitations.

---

> ### Author Response · Authors · 2018-11-25
> **Response to reviewer comments**
>
> Thank you for pointing out such important issues. We will take your input into consideration in a revised manuscript.

---

### Official Review · AnonReviewer3 · 2018-11-07

**Rating:** 3
**Confidence:** 3

**Review:**

This paper proposes an extension of deep image compression model to video compression. The performance is compared with H.264 and MPEG-4.

My main concern is the limited technical novelty and evaluation:
 - The main idea of the architecture is extending 2D convolutions in image compression networks to 3D convolutions, and use skip connections for multi-scale modeling. The 2D to 3D extension is relatively straightforward, and multi-scale modeling is similar to techniques used in, e.g., [Rippel and Bourdev ICML 2017].

 - The reconstruction loss and the entropy loss are commonly used in existing work. One new component is the “temporal consistency loss”. However the impact of the loss is not analyzed in the Experiment section.

 - The evaluation isn’t very extensive. Comparing the proposed method with state-of-the-art codecs (e.g., H.265) or other deep video compression codec (e.g., Wu et al. in ECCV 2018) would be valuable.

 - Since the evaluation dataset is small, evaluation on multiple datasets would make the experiments more convincing.

 - The evaluation is conducted in rather low-bitrate region only (MS-SSIM < 0.9), which is not common point of operation.

 - Finally I agree with AnonReviewer2 on limited description of evaluation details.

Overall I think this paper is not ready for publication yet.

---

> ### Author Response · Authors · 2018-11-25
> **Response to reviewer comments**
>
> We agree that a more in-depth analysis could be performed, particularly regarding the effects of consistency loss, H.265 and Wu et al. ECCV 2018 (although the latter was difficult to be done on time, since it was only very recently published - September 2018). We will take this comments into consideration in a revised manuscript.

---

### Meta-Review · Area_Chair1 · 2018-12-14
**problems with the employed metrics - lack of novelty over static image neural compression**

**Confidence:** 5
**Recommendation:** Reject

**Metareview:**

The paper proposes a neural network architecture for video compression. The reviewers point out lack of novelty with respect to recent neural compression works on static images, which the present paper extends by adding a temporal consistency loss. More importantly, reviewers point our severe problems with the metrics used to measure compression quality, which the authors promise to take into account in a future manuscript.